# AutoPBO: LLM-powered Optimization for Local Search PBO Solvers

## Abstract

Pseudo-Boolean Optimization (PBO) provides a powerful framework for modeling combinatorial problems through pseudo-Boolean (PB) constraints. Local search solvers have shown excellent performance in PBO solving, and their efficiency is highly dependent on their internal heuristics to guide the search. Still, their design often requires significant expert effort and manual tuning in practice. While Large Language Models (LLMs) have demonstrated potential in automating algorithm design, their application to optimizing PBO solvers remains unexplored. In this work, we introduce *AutoPBO*, a novel LLM-powered framework to automatically enhance PBO local search solvers. We conduct experiments on a broad range of four public benchmarks, including one real-world benchmark, a benchmark from PB competition, an integer linear programming optimization benchmark, and a crafted combinatorial benchmark, to evaluate the performance improvement achieved by *AutoPBO* and compare it with six state-of-the-art competitors, including two local search PBO solvers *NuPBO* and *OraSLS*, two complete PB solvers *PBO-IHS* and *RoundingSat*, and two mixed integer programming (MIP) solvers *Gurobi* and *SCIP*. *AutoPBO* demonstrates significant improvements over previous local search approaches, while maintaining competitive performance compared to state-of-the-art competitors. The results suggest that *AutoPBO* offers a promising approach to automating local search solver design.

## 1 Introduction

Pseudo-Boolean Optimization (PBO) plays an important role in solving a wide range of combinatorial problems Boros & Hammer (2002), which seeks an assignment of values to a set of Boolean variables that optimizes a linear objective function under Pseudo-Boolean (PB) constraints. Due to its powerful expressiveness and the convenience to make use of properties of boolean variables, PBO has demonstrated broad applicability across various domains, including VLSI design, economic modeling, computer vision, and manufacturing optimization Wille et al. (2011); Zhang et al. (2011); Roussel & Manquinho (2021).

Along with the wide usages of the PBO problem in industrial and application domains, solving the PBO is then a non-negligible topic. However, the solving of PBO is a challenging task as the problem is NP-hard Buss & Nordström (2021). In previous studies, the solving methods can be divided into two classes: complete methods and incomplete methods. The complete methods solves the problem to optimal and proves the optimality, while incomplete methods do not guarantee to compute the optimal assignment but try to compute good solutions in a short time.

Research on complete methods for solving PBO has developed multiple approaches. First, since PB constraints can be naturally treated as 0-1 linear constraints, mixed-integer programming (MIP) solvers such as *SCIP* Bestuzheva et al. (2021) and *Gurobi* Gurobi Optimization, LLC (2021) can be directly applied to solve PBO problems. Second, by translating PB constraints into conjunctive normal form (CNF), the problem can be solved using SAT solvers based on Conflict-Driven Clause Learning (CDCL), including *MINISAT+* Eén & Sörensson (2006), *Open-WBO* Martins et al. (2014), and *NaPS* Sakai & Nabeshima (2015). Beyond these, advanced methods have been developed for more efficient PBO solving. The cutting planes technique, which goes beyond the resolution power of CDCL, is implemented in solvers like *sat4j* Le Berre & Parrain (2010) and *RoundingSat* Elffers &

Nordström (2018; 2020); Devriendt et al. (2021). Additionally, the implicit hitting set (IHS) method has been successfully adapted to PBO in solvers such as *PBO-IHS* Smirnov et al. (2021; 2022).

Complete algorithms often struggle with large-scale instances, leading to the development of incomplete approaches, among which local search stands out as a representative strategy Lei et al. (2021); Chu et al. (2023b); Zhou et al. (2023). The first notable solver *LS-PBO* Lei et al. (2021) introduced a weighting scheme and a scoring function that jointly handle hard and soft constraints. Several key extensions followed: *DeciLS-PBO* Jiang et al. (2023) enhanced the framework by incorporating unit propagation; *NuPBO* Chu et al. (2023b) proposed enhanced scoring functions and weighting schemes; and *DLS-PBO* Chen et al. (2024) implemented dynamic scoring functions. Additionally, *OraSLS* Iser et al. (2023) utilizes an oracle mechanism to guide the local search approach in PBO.

The efficiency of local search solvers heavily relies on internal heuristics to guide the search process. In the past, many works on designing different algorithmic components such as weighting scheme and score functions have been proposed, as in Thornton (2005); Cai & Su (2013); Cai et al. (2014); Cai & Lei (2020); Lei et al. (2021); Chu et al. (2023a; 2024). However, those works are based on human-designed techniques and designing these heuristics often demands substantial expert effort and manual tuning in practice. On the other hand, recent developments on the LLM-based algorithm design start a new paradigm of algorithm design and show the capability of large language models (LLMs) in automating algorithm design. However, the application of LLMs to building PBO solvers remains unexplored, presenting a promising direction for future research.

Current works on automated algorithm design for combinatorial optimization problems are mainly on designing evolutionary algorithms for specific problems or optimizing heuristics in simple solvers. FunSearch Romera-Paredes et al. (2024) pioneered the integration of pretrained LLMs with evolutionary search, initiating heuristic discovery through iterative code generation. Then, EoH Liu et al. (2024) extends this paradigm through dual-representation evolution, and ReEvo Ye et al. (2024) introduces a structured reflection mechanism to guide evolutionary search. What is more, AutoSAT Sun et al. (2024; 2025) focus on optimizing heuristics in SAT solvers, AlphaEvolve Novikov et al. (2025) pushes the paradigm further by ensembling LLMs with automated evaluators in an evolutionary loop, enabling the discovery of entire algorithmic codebases.

The works on automated heuristic design for the general form problem (i.e. problems with general constraints types, such as Integer Programming, Pseudo Boolean Optimization, etc) is rare, and current approaches of designing heuristics for specific types of problems face critical limitations in this scenario: 1) the general-problem solver usually have complex structure with various algorithm components, and thus results in long-context of source codes, being much more sophisticated than evolutionary algorithms for specific types of problems; 2) A general form problem usually allows a wide range of types of constraints rather than specific types of problems normally has quite limited and known types of constraints, making the heuristics design for these two scenarios quite different. Thus, the algorithm design for the general form optimization problem is still challenging, and to our knowledge, there is no prior work on the LLM-driven automated design for PBO solver.

We consider the automated optimization for local search solvers of the PBO problem. Specifically, we consider to enhance the existing state-of-the-art local search solver for PBO by leveraging the power of LLM. We try to address the above challenges by designing methods from three considerations:1) Enhancing LLMs' comprehension of solver codes with complex structures; 2) Reducing errors or invalid modifications during code generation; 3) Improving the solver's efficiency by optimizing its algorithm as a composition that involves multiple functions.

In this work, we design a novel LLM-powered framework to automatically enhance PBO local search solvers. We propose a multi-agent system integrated with a greedy search strategy, enabling closed-loop, feedback-driven optimization. Furthermore, we introduce a structuralized local search PBO solver *StructPBO*, with a clearer structure of codes, that could be used as an input for automatic optimization frameworks. This design helps LLMs to effectively comprehend and optimize PBO-specific search algorithms and was adopted in our system.

Bring the above ideas together, we design our *AutoPBO* framework to automatically enhancing local search solvers of PBO. Experimental results demonstrate that *AutoPBO* significantly improves the performance of local search PBO solvers, offering a promising approach to automating local search solver design.

## 2 PRELIMINARIES

### 2.1 PSEUDO-BOOLEAN OPTIMIZATION

A linear pseudo-Boolean (PB) constraint is expressed as: $\sum_{j=1}^{n} a_j l_j \rhd b$, where $a_j, b \in \mathbb{Z}$ are integer coefficients, $b$ is the threshold, $\rhd \in \{=, >, \geq, <, \leq\}$ is a relational operator, and each $l_j$ is a literal (either a Boolean variable $x_i$ or its negation $\neg x_i$).

In this work, we assume all PB constraints are in the normalized form $\sum_{j=1}^{n} a_j l_j \geq b$ with $a_j, b \in \mathbb{N}_0^+$ (non-negative integers). This assumption is without loss of generality since all PB constraints can be converted to this form by expressing equalities as inequality pairs and applying the identity $x_i = 1 - \neg x_i$ to ensure non-negative coefficients Roussel & Manquinho (2021).

An assignment $\alpha$ is a mapping from variables to $\{0, 1\}$. A PB constraint $c$ is *satisfied* under $\alpha$ if the inequality $\sum_{j=1}^{n} a_j l_j \geq b$ holds; otherwise, $c$ is *violated*. The *violation degree* of $c$ under $\alpha$, denoted $\text{viol}(c)$, quantifies how far $c$ is from being satisfied: $\text{viol}(c) = \max\left(0, b - \sum_{j=1}^{n} a_j l_j\right)$, which is zero if $c$ is satisfied, and otherwise measures the shortfall. A *PB formula $F$* is a conjunction of PB constraints, and an assignment that satisfies all constraints in $F$ is called a *feasible solution*.

A pseudo-Boolean optimization (PBO) instance consists of a PB formula $F$ together with a linear Boolean objective function $\sum_{j=1}^{n} e_j l_j + d$, where $e_j \in \mathbb{N}^+$ and $d \in \mathbb{Z}$. Since all PB constraints must hold, they are treated as *hard constraints*. For any assignment $\alpha$, its objective value is denoted as $obj(\alpha)$. A feasible solution $\alpha_1$ is considered superior to another solution $\alpha_2$ if $obj(\alpha_1) < obj(\alpha_2)$. The objective of PBO is to identify a feasible assignment $\alpha$ that minimizes $obj(\alpha)$.

### 2.2 LOCAL SEARCH FOR PBO

The local search is a general algorithmic paradigm fo solving combinatorial optimization problems. It typically begins with an initial solution and iteratively explores the neighborhood of the current solution, seeking an improved candidate solution. If such a solution is found, it replaces the current one; otherwise, the search either terminates or employs strategies to escape local optima. The process continues until a stopping criterion is met, such as reaching a maximum number of iterations, a time limit, or a satisfactory solution quality threshold.

In a typical local search process of PBO, given an instance $F$, the local search algorithm starts from an initial solution $\alpha$, then iteratively modifies $\alpha$ by selecting variables heuristically and applying corresponding **operators** (e.g., the $flip$ operator in pseudo-Boolean optimization) until a feasible solution is found. An operation is obtained when an operator is specified with a variable, and it is easy to see there could be multiple operations for generating a new solution. During this process, **scoring functions** evaluate candidate operations, prioritizing operations that are likely to improve solution quality. An important factor normally included in scoring functions is the weights of constraints. A **weighting scheme** is adopted to compute the weights of constraints in the scoring function, representing the importance of the constraints.

In general, for greedy variable flipping, PBO local search (LS) algorithms employ a scoring function integrated with the weights of constraints. Let $w(c)$ denote the weight of a hard constraint $c$, and $w(o)$ denote the weight of the objective function $o$. For instance, in *LS-PBO*, the penalty for a constraint $c$ is defined as $penalty(c) = w(c) \times viol(c)$, and the penalty for the objective function under the current assignment $\alpha$ is $penalty(o) = w(o) \times obj(\alpha)$. In *NuPBO*, a smoothed penalty method is proposed to balance the $viol$ values across constraints. Specifically, for a hard constraint $c$, the penalty function is redefined as: $penalty(c) = \frac{w(c) \times viol(c)}{smooth(c)}$ where $smooth(c)$ represents a smoothing coefficient derived from constraint properties. Similarly, for the objective function $o$, the penalty is adjusted to: $penalty(o) = \frac{w(o) \times obj(\alpha)}{smooth(o)}$; with $smooth(o)$ often calculated as the average of the objective function's coefficients. Across these algorithms, the hard score $hscore(x)$ of a variable $x$ quantifies the reduction in the total penalty of all hard constraints when $x$ is flipped. The soft score $oscore(x)$ measures the reduction in the objective function's penalty after flipping $x$. The scoring function of $x$ is defined as $score(x) = hscore(x) + oscore(x)$, a linear combination of the hard and soft scores.

# 3 THE AUTOMATED OPTIMIZATION FRAMEWORK

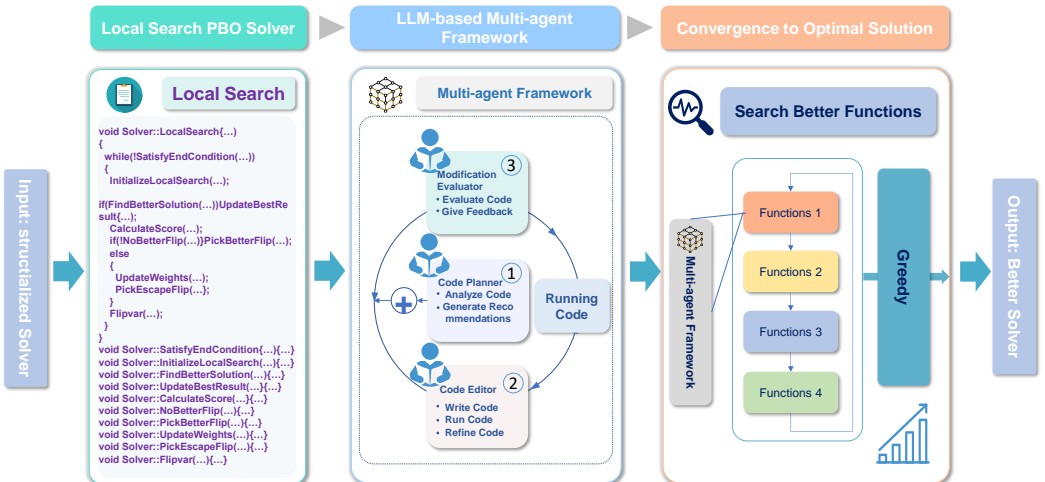

Figure 1: Architecture of AutoPBO. With a structuralized local search PBO solver as input, AutoPBO implements greedy strategy to optimize heuristic functions iteratively. For each iteration, AutoPBO employs the original solver code to instantaneously create an understanding and recommendations for heuristic functions, subsequently engages in code generation and performance assessment by three agents. Upon completion, AutoPBO returns the optimal solver code.

For automating the optimization of codes for local search solver of PBO, we propose a LLM-based multi-agent framework, named *AutoPBO*. Our framework is based on three types of agents and a greedy-based iterative method to achieve a better performance solver.

As illustrated in Figure 1, our framework begins by loading a local search PBO solver, then a number of iterative code optimizing rounds are launched. In each round, we perform several distinct and independent code modifications work, each modification work is realized by the three LLM agents collaboratively. After an optimization round, several versions of code are generated and then a greedy-based selection strategy is applied to select the most effective version for the next iteration. By repeating, the framework ultimately generates the resulting solver.

## 3.1 OPTIMIZATION BY MULTIPLE AGENTS

There are three specialized LLM agents in our framework, each of which has its own functionality and distinct roles, i.e., the Code Optimization Planner, Code Editor, and Modification Evaluator. Each of them serves different functionalities in our framework as follows:

- Code Optimization Planner: Analyze key code segments to recognize the function targeted for modification and generate modification plans.
- Code Editor: Realize the code modification to improve the code, following the advice generated by the Code Optimization Planner.
- Modification Evaluator: Evaluate the modified code and generate advice for future improvement.

Those three agents operate through an automated cycle of modification plan generation, code edit iteration, and dynamic evaluation.

A code optimization round consists two phases: the planning phase and the editing phase. In the planning phase, the Code Optimization Planner Agent identifies possible optimization ways and generates preliminary improvement plans, such as "Dynamic Score Ratios: Adjust h_score_ratio and s_score_ratio dynamically based on the current state of the search (e.g., increase the weight of hard constraints when the solution is infeasible", "Include a term that considers the age of the variable

(time since last flip) to encourage diversification and escape local optima", etc. The details of the implementations of the Code Optimization Planner is presented in Section 4.2.

The editing phase is built by interactions of the Code Editor and the Modification Evaluator, for multiple times. The Code Editor Agent performs the actual work of code modifications and runs for multiple times in the editing phase. Initially, it modifies the code modifications with the optimization plans provided by the Code Optimization Planner as input, and generates the first version of the code. It identifies if the modification has actual significant meanings, by excluding trivial modifications such as variable re-naming, parameter changing, etc. An actual compilation and running of the code is also performed at this time, to collect information such as the existence of compilation errors or running information.

The running information and the evaluation results from the Modification Evaluator will then be sent to the code Editor as feedback for the next modification. This interaction will be performed by the Code Editor, Modification Evaluator several times, resulting in a final version of code for the current code optimization round.

In the following, we will first present the implementation of different agents, then introduce the greedy-based iterative methods on top of code optimization rounds, which jointly form our whole optimization framework for PBO local search solver.

## 3.2 PROMPT ENGINEERING

| Agent | Tasks | Tips |
|-------|-------|------|
| Code Optimization Planner | 1. Comprehensively analyzing all key code segments
2. Proposing possible modification manners | 1. Complete review of all relevant code sections
2. Generation of both practical and theoretically promising proposals
3. Strict JSON format adherence |
| Code Editor | Phase 1:
1. Precise interpretation of Planner's description
2. Directional suggestions for code changes
Phase 2:
1. Analyze experimental outcomes
2. Produce superior code versions | Phase 1:
1. Preserve original function signatures
2. Prevent undefined variable introduction
3. Guarantee differentiation from previous codes
Phase 2:
1. Maintain code formatting
2. Avoid redundant modifications |
| Code Modification Evaluator | 1. Evaluation and classification of outputs | 1. Thorough syntax verification
2. Comparative analysis for meaningful improvements
3. Categorical classification |

Table 1: Agent Tasks and Tips Overview

The three agents are implemented by different prompt, which are designed following the principles proposed by OpenAI's foundational guidelinesOpenAI (2023), we have developed a structured prompt to implement different functionalities for them.

Our prompt includes A *Role* set to the agent, the *Tasks* definitions, the *Tips* to provide suggestions and the complete PBO solver code appended at the end to ensure all agents share the same context. All three agents share a common *Role* configuration as a solver researcher attempting to improve the heuristics in a PBO solver. This foundational *Role* description establishes the professional identity and overarching objective for each agent in the optimization process. *Tasks* and *Tips* of every agent are shown in Table 1.

## 3.3 CONVERGENCE TO OPTIMAL SOLUTION

As illustrated in Figure 1, we implement an iterative greedy algorithm that progressively constructs the improved solver through three steps:

- **Code Optimization Round**: Agents optimize one function at a time, while keeping others functions fixed. For example, we first generate multiple optimized versions of the UpdateWeights function through LLM agents, then select the best-performing version before proceeding to optimize CalculateScore. Implementation Process is that the LLM agents first generate multiple optimized versions of the function as illustrated in 3.1. Then our

framework automates the following workflow: it constructs solver instances for each function version, runs these solvers in parallel on certain dataset, and automatically collects their outputs—including feasibility status (Feasible or Infeasible) and objective values($obj$) across all test instances. Finally, the framework determines the best-performing version through automatic comparison by tallying the total feasible solutions (Feasible count) and the number of instances where a solver's $obj$ outperforms *StructPBO* (Win count), selecting the version that achieves the highest count on both metrics.

- **Modification Propagating**: The selected optimal function (e.g., improved UpdateWeights) is immediately integrated into *StructPBO*. Subsequent optimizations (e.g., CalculateScore refinement) are then performed on this updated *StructPBO*.

- **Iterative Improvement**: This process repeats until all target functions are optimized.

This approach addresses the inherent dependency between functions. Consider the interaction between UpdateWeights and CalculateScore: The scoring function in CalculateScore must reflect the latest clause weights from UpdateWeights to properly prioritize constraint satisfaction. If we independently optimize both functions and combine their best versions, the scoring mechanism might ignore updated weight distributions, leading to inconsistent optimization behavior. Our greedy strategy prevents such conflicts by enforcing sequential adaptation - the improved CalculateScore automatically adapts to the newly integrated UpdateWeights through Modification Propagating.

Together with these three steps and the three agents involved, we iteratively improve our code, trying to get a better performance solver.

## 3.4 A New Structuralized Local Search PBO Solver: *StructPBO*

As we mentioned in Section 1, previous studies on automated algorithm design usually focus on combinatorial optimization problems with known types and simple-architecture algorithms, which are quite different from general problem solvers with complex structures. Notably, state-of-the-art Local Search PBO solvers typically involve advanced programming techniques and multiple algorithmic components to achieve high efficiency. This makes it challenging for LLMs to directly process or optimize such solvers.

In our preliminary experiments, using LLMs to generate a PBO solver from scratch under the EoH mechanism resulted in only simplistic local search algorithms with basic scoring functions and random perturbations (see Appendix C). Moreover, when attempting to optimize existing Local Search PBO solvers through established solver optimization frameworks, the high complexity and tight coupling of the codebase led to frequent syntactic errors and logical inconsistencies in LLM-generated code.

To address these issues, we propose a predefined, structured Local Search PBO framework named *StructPBO*. Aligning with the basic components of local search, we define the major functionalities of a solver and build a structuralized solver *StructPBO*, following the design of the state-of-the-art *NuPBO* Chu et al. (2023b).

The overall workflow of *StructPBO* is summarized in Algorithm 1, where the solver iteratively maintains a complete assignment, updates the best solution, and employs a heuristic-driven variable selection mechanism. The key idea is to balance constraint satisfaction and objective optimization through a composite scoring function, where dynamic penalty weights adaptively shift the search focus according to the current search status. This enables effective navigation of the search space while avoiding stagnation in local optima.

Implementation details, including the line-by-line description of Algorithm 1, the formulation of scoring functions, the weight adaptation strategy, and the modular decomposition into seven independent functions, are provided in Appendix C.

## 4 Experiments

In this section, we introduce the experimental settings and present extensive experiments on 4 PBO benchmarks. First, we evaluate the effectiveness of *AutoPBO* as a framework for enhancing baseline

---

**Algorithm 1:** the structPBO solver

---

**Input:** PBO instance $F$, cutoff time $cutoff$.
**Output:** The best solution $\alpha^*$ found and its objective function value $obj^*$, or "No solution
 found".

1 $\alpha^* := \varnothing, \quad obj^* := +\infty$;
2 $\alpha :=$ InitializeAssignment();
3 **while** *elapsed time $<$ cutoff* **do**
4  **if** *$\alpha$ is feasible **and** $obj(\alpha) < obj^*$* **then**
   // Update best solution and its objective function value
5   $\alpha^* := \alpha, \quad obj^* := obj(\alpha)$;
6  **for** *each variable $x$* **do**
7   hscore$(x) := \Delta\text{Penalty}_{\text{hard}}(x)$;
8   oscore$(x) := \Delta\text{Penalty}_{\text{obj}}(x)$;
9   score$(x) :=$ CalculateScore(hscore$(x)$, oscore$(x)$);
10  **if** $D := \{x | Score(x) > 0\} \neq \varnothing$ **then**
   // A variable is picked accordingly
11   $x :=$ PickBestVariable($D$);
12  **else**
   // Stuck in a local optimum
13   UpdateWeights(F);
   // A variable is picked according to local-optima-escaping
   heuristics
14   $x :=$ PickEscapeVariable($F$);
15  $\alpha := \alpha$ with $x$ flipped;
16 **if** $\alpha^* \neq \varnothing$ **then return** $\alpha^*$ and $obj^*$;
17 **else return** No solution found;

---

solvers, demonstrating that it consistently improves performance across multiple benchmarks. Second, we compare *AutoPBO* with state-of-the-art PBO solvers to highlight its strong competitiveness. Finally, in Appendix A we provide variance analysis through repeated experiments to demonstrate the statistical stability of *AutoPBO*'s performance.

## 4.1 SETTINGS

### 4.1.1 ENVIRONMENT

Our PBO solver is implemented in C++, while the interface for interacting with LLMs is developed in Python. The solver is compiled using g++ 9.4.0. All experiments are performed on an Ubuntu 20.04.4 LTS server, which is equipped with two AMD EPYC 7763 processors. Each processor operates at a base frequency of 2.45 GHz. The server is configured with 1TB of RAM. For all experiments, the DeepSeek[1] LLM is employed.

### 4.1.2 BENCHMARKS AND DATASETS

We evaluate *AutoPBO* on four widely-used PBO benchmarks: PB16, MIPLIB, CRAFT, and Real-world, comprising 47 datasets in total. All benchmarks are randomly split into training and testing sets with a 1:1 ratio. Training sets are used to generate optimized solvers, while testing sets are reserved for evaluation. Full dataset descriptions and sources are provided in Appendix D.1.

### 4.1.3 STATE-OF-THE-ART COMPETITORS

We compare *AutoPBO* with 6 state-of-the-art solvers, including 2 incomplete solver (*i.e., NuPBO* and *OraSLS*) and 4 complete solvers. The 4 complete solvers include 2 PB solvers (*i.e., PBO-IHS*

---

[1]We utilize the DeepSeek-R1 as default in this paper.

| Benchmark | StructPBO | | AutoPBO | |
|---|---|---|---|---|
| | #win | avg_score | #win | avg_score |
| Real-world | 17 | 0.9962 | **29** | **0.9998** |
| CRAFT | 488 | 0.9472 | **513** | **0.9474** |
| MIPLIB | 101 | 0.8450 | **112** | **0.8598** |
| PB16 | 697 | 0.8272 | **775** | **0.8447** |
| Total | 1303 | 0.8738 | **1429** | **0.8849** |

Table 2: *AutoPBO* vs *StructPBO* Performance Comparison (By Benchmark)

| | Gurobi | | SCIP | | PBO-IHS-Tuned | | RoundingSat-Tuned | | OraSLS-Tuned | | NuPBO-Tuned | | AutoPBO | |
|---|---|---|---|---|---|---|---|---|---|---|---|---|---|---|
| Benchmark | #win | avg_score | #win | avg_score | #win | avg_score | #win | avg_score | #win | avg_score | #win | avg_score | #win | avg_score |
| Real-world | 3 | 0.5050 | 0 | 0.1417 | 0 | 0.3311 | 4 | 0.4508 | 2 | 0.3221 | 16 | 0.9649 | **23** | **0.9671** |
| CRAFT | **479** | **0.9783** | 289 | 0.8337 | 277 | 0.7959 | 451 | 0.9690 | 406 | 0.9620 | 454 | 0.9447 | 460 | 0.9449 |
| MIPLIB | **101** | **0.8378** | 62 | 0.5653 | 55 | 0.7044 | 48 | 0.7616 | 52 | 0.7246 | 71 | 0.8212 | 70 | 0.8347 |
| PB16 | **626** | **0.8456** | 314 | 0.6115 | 463 | 0.7537 | 428 | 0.7135 | 464 | 0.8016 | 548 | 0.8181 | 572 | 0.8195 |
| Total | **1209** | **0.8834** | 665 | 0.6740 | 795 | 0.7548 | 931 | 0.8004 | 924 | 0.8398 | 1089 | 0.8653 | 1125 | 0.8674 |

Table 3: Multi-Solver Performance Comparison (By Benchmark)

and *RoundingSat*) and 2 MIP solvers (*i.e., Gurobi* and *SCIP*).Detailed solver information is listed in Appendix D.2.

### 4.1.4 PERFORMANCE METRICS

In our experiments, *AutoPBO* first generates optimization strategies on the training set with a 60-second cutoff time. Then, each solver performs one run within a given cutoff time (300 seconds) on every instance in the testing set for evaluation. We record the cost of the best solution found by solver $S_j$ on instance $I_k$, denoted as $sol_{S_j I_k}$. The cost of the best solution found among all solvers in the same table on instance $I_k$ is denoted as $best_{I_k}$.

Following previous research on PBO, we measure the performance of each solver using two metrics:

- $\#win$: the number of instances where the corresponding $best_{I_k}$ can be obtained by solver $S$ on $B_i$ (*i.e.,* the number of winning instances).

- $avg\_score$: in our experiments, the competition score of solver $S_j$ on instance $I_k$ is represented by $score_{S_j I_k} = \frac{best_{I_k}+1}{sol_{S_j I_k}+1}$, which measures the gap between $sol_{S_j I_k}$ and $best_{I_k}$. If solver $S_j$ could not report a solution on instance $I_k$, then $score_{S_j I_k} = 0$. We use $avg\_score$ to denote the average competition score of a solver on a dataset.

For each of the above two metrics, if a solver obtains a larger metric value on a dataset, then the solver exhibits better performance on the dataset. The results highlighted in **bold** indicate the best performance for the corresponding metric.

### 4.2 RESULTS

#### 4.2.1 IMPROVEMENTS OF LOCAL SEARCH PBO SOLVER

We first evaluate *AutoPBO* on top of *StructPBO* across 47 datasets from 4 benchmarks. Figures 2 and 3 present the datasets with changes in $\#win$ and $avg\_score$ respectively along with the magnitude of these changes. *AutoPBO* demonstrates consistent performance gains: improving $\#win$ on 24 datasets and $avg\_score$ on 23 datasets, with only a single case of marginal degradation in $avg\_score$. This consistent non-negative trend confirms that *AutoPBO* not only avoids harming performance but also delivers significant gains on nearly half of the datasets across all benchmarks.

Table 2 provides a benchmark-level comparison between *AutoPBO* and *StructPBO* in terms of both $\#win$ and $score$. Across all 4 benchmarks, *AutoPBO* demonstrates consistent improvements, raising the total number of $\#win$ and increasing the overall $avg\_score$.

These benchmark-level results highlight the generality and robustness of *AutoPBO*.

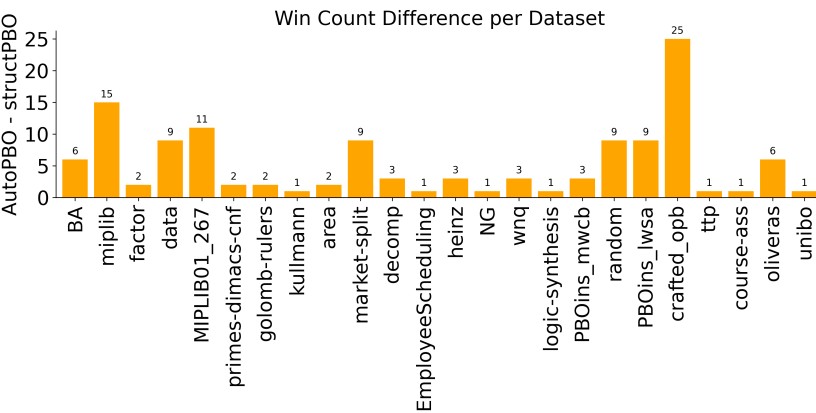

Figure 2: #*win* differences between *AutoPBO* and *StructPBO*. Each bar represents *AutoPBO*'s #*win* minus *StructPBO*'s #*win*. Positive values (orange bars) indicate *AutoPBO*'s advantage.

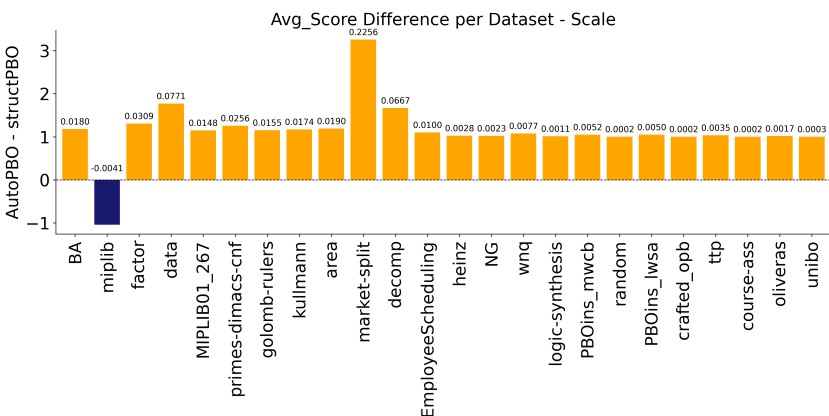

Figure 3: $avg\_score$ differences between *AutoPBO* and *StructPBO*. Each bar represents *AutoPBO*'s $avg\_score$ minus *StructPBO*'s $avg\_score$ (*AutoPBO* − *StructPBO*). Positive values (orange bars) indicate *AutoPBO*'s higher $avg\_score$, while negative values (dark-blue bars) indicate *StructPBO*'s higher $avg\_score$ (after non-linear scaling for better visibility).

### 4.2.2 COMPETITIVE RESULTS OF *AutoPBO*

We conduct a comprehensive comparison between *AutoPBO* and state-of-the-art solvers, as shown in Table 3. To ensure fair comparison, we performed parameter tuning for *PBO-IHS*, *RoundingSat*, *OraSLS*, and *NuPBO* on each dataset. The tuning scripts and final parameter configurations are documented in Code & Data Appendix. The experimental results demonstrate that *AutoPBO* outperforms all open-source solvers in both #*win* and $avg\_score$. Notably, *AutoPBO* shows competitive performance compared to the commercial solver *Gurobi*.

At the dataset level (see Appendix A), *AutoPBO* achieves the highest #*win* on 31 out of 47 datasets and obtains the best $avg\_score$ on 32 datasets. The complete per-dataset comparison reveals that *AutoPBO* consistently delivers robust performance across diverse problem types.

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

## A    THE USE OF LARGE LANGUAGE MODELS

We employed a large language model (LLM) to assist with proofreading and polishing the writing of this paper.

## B    IMPLEMENTATION DETAILS OF *StructPBO*

In this appendix, we provide the detailed implementation of our structuralized local search solver *StructPBO*. While the main text focuses on the overall design and algorithmic mechanism, here we describe the technical components that enable the solver to operate effectively. In particular, we (i) present a line-by-line explanation of Algorithm 1, (ii) define the scoring functions and dynamic weight adaptation strategy, and (iii) illustrate the modular decomposition into seven independently implemented functions.

### B.1    ALGORITHM WALKTHROUGH

Algorithm 1 outlines the search routine of *StructPBO*. The solver maintains a complete assignment at all times and iteratively explores the neighborhood by flipping candidate variables. During each iteration:

- Lines 2–5: Initialize and update the best solution found.
- Lines 6–9: Evaluate candidate flips using the composite score combining $hscore(x)$ and $oscore(x)$.
- Lines 10–12: Select the best candidate based on the composite score.
- Line 13: If the solver is trapped in local optima, trigger weight adaptation to escape stagnation.
- Lines 14–15: Apply the chosen flip to update the assignment.
- Lines 16–17: Terminate when the cutoff or convergence condition is met and return the best solution.

### B.2    SCORING FUNCTIONS

The evaluation of candidate flips relies on two penalty-based scoring functions:

- $hscore(x)$: quantifies the penalty reduction with respect to unsatisfied hard constraints.
- $oscore(x)$: quantifies the penalty reduction for the objective function.

The overall score is a weighted combination:

$$\text{score}(x) = \alpha \cdot hscore(x) + \beta \cdot oscore(x),$$

where weights $\alpha$ and $\beta$ dynamically change according to the search phase. During feasibility-seeking phases, $\alpha \gg \beta$; as the search approaches optimality, $\beta$ increases in relative importance.

### B.3    DYNAMIC WEIGHT ADAPTATION

To avoid stagnation in local optima, *StructPBO* periodically adjusts the weights $\alpha$ and $\beta$. This adaptation allows the solver to balance between repairing constraint violations and improving the objective. The adaptation strategy follows the principle that hard constraints must be prioritized initially, but once feasibility is stable, the solver progressively emphasizes objective optimization.

### B.4    MODULAR IMPLEMENTATION

To facilitate LLM-based optimization and reduce code complexity, *StructPBO* decomposes the solver into seven independently implemented functions. Each function corresponds to a key component of the local search process:

- **InitializeAssignment**: Heuristically generates an initial complete assignment.

- **Penalty_hard**: Heuristically computes the hard constraint penalty reduction.

- **Penalty_obj**: Heuristically computes the objective penalty reduction.

- **CalculateScore**: Combines hard and soft penalties into a dynamic composite score.

- **PickBestVariable**: Selects the most promising variable based on the composite score.

- **UpdateWeights**: Adjusts the weights of hard and objective penalties dynamically during the search.

- **PickEscapeVariable**: Identifies variables for diversification when stagnation is detected.

This modular structure allows isolated function-level improvements without breaking the overall consistency of the solver, and provides a clean interface for integration with automated solver optimization frameworks. Detailed descriptions of each function's inputs, outputs, and internal logic are provided in the code appendix.

## C   INDEPENDENT FUNCTIONS IN STRUCTPBO

- **InitializeAssignment**: Heuristically generates an initial complete assignment

- **Penalty_hard**: Heuristically computes the hard constraint penalty reduction

- **Penalty_obj**: Heuristically computes the objective penalty reduction

- **CalculateScore**: Heuristically combines hard and soft penalties into a dynamic composite score

- **PickBestVariable**: Heuristically selects the most promising variable

- **UpdateWeights**: Heuristically selects the most promising variable

- **PickEscapeVariable** : Heuristically identifies variables for diversification when stagnation is detected

## D   DETAILED EXPERIMENTAL SETTINGS

### D.1   BENCHMARKS AND DATASETS

To comprehensively evaluate the effectiveness of *AutoPBO*, we employ four widely-used PBO benchmark families. Detailed descriptions are provided below.

- `PB16`: The OPT-SMALLINT-LIN benchmark from the 2016 pseudo-Boolean competition, comprising 1600 diverse instances from multiple categories.[2] Following the categorization in (Smirnov et al., 2021), we divide PB16 into 42 datasets based on their applications.

- `MIPLIB`: A benchmark of 0-1 integer linear programming problems, containing 267 instances of various types, as introduced in (Devriendt et al., 2021).[3]

- `CRAFT`: A collection of 1025 crafted combinatorial problems with small integer coefficients, also from (Devriendt et al., 2021).[4]

- `Real-world`: A benchmark set containing three application-driven problems: the Minimum-Width Confidence Band Problem (MWCB, 24 instances), the Seating Arrangements Problem (SAP, 21 instances), and the Wireless Sensor Network Optimization Problem (WSNO, 18 instances). All problem descriptions, encodings, and instances are from (Lei et al., 2021).[5]

---

[2]http://www.cril.univ-artois.fr/PB16/bench/PB16-used.tar
[3]https://zenodo.org/record/3870965
[4]https://zenodo.org/record/4036016
[5]https://lcs.ios.ac.cn/%7ecaisw/Resource/LS-PBO/

## D.2 STATE-OF-THE-ART COMPETITORS

To provide a comprehensive comparison, we include six state-of-the-art solvers in our experiments. Their information is detailed below:

- *NuPBO* (Chu et al., 2023b): The state-of-the-art local search solver for solving PBO.
- *OraSLS* (Iser et al., 2023): a recent oracle-based SLS algorithm for PBO, which improves upon previous pure SLS approaches.
- *PBO-IHS* (Smirnov et al., 2022): A PBO solver that utilizes the implicit hitting set approach and building upon *RoundingSat* (Elffers & Nordström, 2018).
- *RoundingSat* (Devriendt et al., 2021): A PBO solver combining core-guided search with cutting planes reasoning.
- *Gurobi* (Gurobi Optimization, LLC, 2021): One of the most powerful commercial MIP solvers (Version 12.0.2). The default configuration is used, along with a single thread.
- *SCIP* (Gamrath et al., 2020): One of the fastest non-commercial solvers for MIP (Version 8.0.4).

# E EXTENDED EXPERIMENTAL RESULTS

## E.1 DETAILED EXPERIMENTAL RESULTS

Tables 4 and 5 present the detailed per-dataset evaluation results of *AutoPBO* across all 47 datasets, extending the benchmark-level analysis in Section 5.2. These comprehensive results demonstrate that *AutoPBO* consistently outperforms *StructPBO* while maintaining competitive performance against state-of-the-art solvers. Specifically, *AutoPBO* achieves superior performance in both $\#win$ and $avg\_score$ metrics across diverse problem types, further validating its robust performance.

| Dataset | *StructPBO* #*win* | *AutoPBO* #*win* | *StructPBO* $avg\_score$ | *AutoPBO* $avg\_score$ |
|---|---|---|---|---|
| BA | 5 | **11 (+6)** | 0.9820 | **1.0000 (+0.0180)** |
| EmployeeSche | 8 | **9 (+1)** | 0.8789 | **0.8889 (+0.0100)** |
| MIPLIB01_267 | 101 | **112 (+11)** | 0.8450 | **0.8598 (+0.0148)** |
| NG | 14 | **15 (+1)** | 0.9977 | **1.0000 (+0.0023)** |
| PBOins_lwsa | 1 | **10 (+9)** | 0.9949 | **0.9999 (+0.0050)** |
| PBOins_mwcb | 7 | **10 (+3)** | 0.9945 | **0.9997 (+0.0052)** |
| PBOins_wsno | **9** | **9** | **1.0000** | **1.0000** |
| area | 13 | **15 (+2)** | 0.9810 | **1.0000 (+0.0190)** |
| areaDelay | **15** | **15** | **1.0000** | **1.0000** |
| bounded_golo | **9** | **9** | **0.4444** | **0.4444** |
| course-ass | 2 | **3 (+1)** | 0.9998 | **1.0000 (+0.0002)** |
| crafted_opb | 488 | **513 (+25)** | 0.9472 | **0.9474 (+0.0002)** |
| cudf | **11** | **11** | **0.5455** | **0.5455** |
| data | 30 | **39 (+9)** | 0.4452 | **0.5223 (+0.0771)** |
| decomp | 2 | **5 (+3)** | 0.9333 | **1.0000 (+0.0667)** |
| domset | **8** | **8** | **1.0000** | **1.0000** |
| dt-problems | **30** | **30** | **1.0000** | **1.0000** |
| factor | 86 | **88 (+2)** | 0.8756 | **0.9066 (+0.0310)** |
| fctp | **16** | **16** | **0.1250** | **0.1250** |
| featureSubsc | **10** | **10** | **1.0000** | **1.0000** |
| flexray | **5** | **5** | **0.4000** | **0.4000** |
| frb | **20** | **20** | **1.0000** | **1.0000** |
| garden | **4** | **4** | **1.0000** | **1.0000** |
| golomb-ruler | 6 | **8 (+2)** | 0.6433 | **0.6589 (+0.0156)** |
| graca | **10** | **10** | **1.0000** | **1.0000** |
| haplotype | **4** | **4** | **1.0000** | **1.0000** |
| heinz | 20 | **23 (+3)** | 0.6494 | **0.6522 (+0.0028)** |
| kullmann | 3 | **4 (+1)** | 0.9826 | **1.0000 (+0.0174)** |
| logic-synthe | 36 | **37 (+1)** | 0.9989 | **1.0000 (+0.0011)** |
| market-split | 11 | **20 (+9)** | 0.3244 | **0.5500 (+0.2256)** |
| milp | **17** | **17** | **0.7059** | **0.7059** |
| minlplib | **49** | **49** | **1.0000** | **1.0000** |
| miplib | 36 | **51 (+15)** | **0.7706** | 0.7665 (-0.0041) |
| mps | **2** | **2** | **1.0000** | **1.0000** |
| oliveras | 57 | **63 (+6)** | 0.9975 | **0.9992 (+0.0017)** |
| pbfvmc-formu | **11** | **11** | **1.0000** | **1.0000** |
| poldner | **3** | **3** | **1.0000** | **1.0000** |
| primes-dimac | 75 | **77 (+2)** | 0.8077 | **0.8332 (+0.0255)** |
| radar | **6** | **6** | **1.0000** | **1.0000** |
| random | 13 | **22 (+9)** | 0.7725 | **0.7727 (+0.0002)** |
| routing | **8** | **8** | **1.0000** | **1.0000** |
| synthesis-pt | **5** | **5** | **1.0000** | **1.0000** |
| trarea_ac | **5** | **5** | **1.0000** | **1.0000** |
| ttp | 3 | **4 (+1)** | 0.9965 | **1.0000 (+0.0035)** |
| unibo | 17 | **18 (+1)** | 0.3886 | **0.3889 (+0.0003)** |
| vtxcov | **8** | **8** | **1.0000** | **1.0000** |
| wnq | 4 | **7 (+3)** | 0.9895 | **0.9971 (+0.0076)** |
| Total | 1303 | **1429 (+126)** | 0.8738 | **0.8849 (+0.0110)** |

Table 4: *AutoPBO* vs *StructPBO* Performance Comparison (By Dataset)

| Dataset | Gurobi #win | avg_score | SCIP #win | avg_score | PBO-IHS-Tuned #win | avg_score | RoundingSat-Tuned #win | avg_score | OraSLS-Tuned #win | avg_score | NuPBO-Tuned #win | avg_score | AutoPBO #win | avg_score |
|---|---|---|---|---|---|---|---|---|---|---|---|---|---|---|
| BA | **12** | **0.9984** | 0 | 0.2549 | 0 | 0.8292 | 3 | 0.8971 | 0 | 0.8769 | 0 | 0.9665 | 3 | 0.9819 |
| EmployeeSc | 6 | 0.7579 | 0 | 0.0000 | 1 | 0.5786 | 5 | 0.8090 | 5 | 0.8195 | 7 | 0.8789 | **8** | **0.8889** |
| MIPLIB01_2 | **101** | **0.8378** | 62 | 0.5653 | 55 | 0.7044 | 48 | 0.7616 | 52 | 0.7246 | 71 | 0.8212 | 70 | 0.8347 |
| NG | 11 | 0.9466 | 0 | 0.0000 | 0 | 0.8847 | 2 | 0.9405 | 0 | 0.9176 | 7 | **0.9847** | 7 | **0.9847** |
| PBOins_lws | 0 | 0.0811 | 0 | 0.0000 | 0 | 0.0000 | 0 | 0.0000 | 0 | 0.0000 | 3 | 0.9957 | **9** | **0.9997** |
| PBOins_mwc | 2 | 0.7366 | 0 | 0.1080 | 0 | 0.5599 | 3 | 0.6587 | 0 | 0.2685 | 6 | 0.9133 | **7** | **0.9153** |
| PBOins_wsn | 1 | 0.7144 | 0 | 0.3599 | 0 | 0.4305 | 1 | 0.7245 | 2 | 0.7871 | **7** | **0.9962** | **7** | **0.9962** |
| area | **15** | **1.0000** | 8 | 0.8974 | 10 | 0.9265 | 14 | 0.9892 | 10 | 0.9430 | 13 | 0.9769 | 14 | 0.9949 |
| areaDelay | **15** | **1.0000** | 0 | 0.8691 | 8 | 0.9690 | **15** | **1.0000** | 6 | 0.9683 | **15** | **1.0000** | **15** | **1.0000** |
| bounded_go | 3 | 0.5215 | 1 | 0.2753 | 2 | 0.6140 | 3 | 0.6544 | **5** | **0.6610** | 1 | 0.2561 | 1 | 0.3172 |
| course-ass | **3** | **1.0000** | 1 | 0.7863 | 1 | 0.9871 | 2 | 0.9998 | 2 | 0.9976 | 2 | 0.9998 | **3** | **1.0000** |
| crafted_op | **479** | **0.9783** | 289 | 0.8337 | 277 | 0.7959 | 451 | 0.9690 | 406 | 0.9620 | 454 | 0.9447 | 460 | 0.9449 |
| cudf | 6 | 0.5455 | 6 | 0.5455 | **10** | **0.9987** | 6 | 0.5455 | 4 | 0.3636 | 6 | 0.5455 | 6 | 0.5455 |
| data | **41** | **0.6512** | 20 | 0.2555 | 14 | 0.3271 | 16 | 0.3640 | 19 | 0.4966 | 16 | 0.4245 | 17 | 0.4376 |
| decomp | 1 | 0.7381 | 0 | 0.7788 | 0 | 0.3150 | 0 | 0.0000 | 0 | 0.9355 | 1 | 0.9206 | **5** | **1.0000** |
| domset | 1 | 0.9872 | 0 | 0.9035 | 0 | 0.9774 | 0 | 0.9375 | 0 | 0.9269 | 7 | 0.9993 | **8** | **1.0000** |
| dt-problem | **30** | **1.0000** | **30** | **1.0000** | **30** | **1.0000** | **30** | **1.0000** | 24 | 0.8000 | **30** | **1.0000** | **30** | **1.0000** |
| factor | **96** | **0.9583** | **96** | **0.9583** | **96** | **0.9583** | **96** | **0.9583** | **96** | **0.9583** | **96** | **0.9583** | 87 | 0.9006 |
| fctp | **2** | **0.1250** | 1 | 0.1211 | 0 | 0.0000 | 0 | 0.0000 | **2** | **0.1250** | **2** | **0.1250** | **2** | **0.1250** |
| featureSub | 3 | 0.9869 | 0 | 0.7037 | **10** | **1.0000** | **10** | **1.0000** | **10** | **1.0000** | **10** | **1.0000** | **10** | **1.0000** |
| flexray | **2** | **0.4000** | **2** | **0.4000** | **2** | **0.4000** | **2** | **0.4000** | 0 | 0.0000 | **2** | **0.4000** | **2** | **0.4000** |
| frb | 2 | 0.9966 | 0 | 0.9880 | 0 | 0.9942 | 0 | 0.9923 | 0 | 0.9922 | 20 | **1.0000** | 20 | **1.0000** |
| garden | 3 | 0.9057 | 1 | 0.8628 | 3 | 0.7500 | 2 | 0.9256 | 2 | 0.9050 | **4** | **1.0000** | **4** | **1.0000** |
| golomb-rul | 6 | 0.6584 | 5 | 0.6171 | **8** | **0.8466** | 1 | 0.0000 | 7 | 0.6667 | 4 | 0.5686 | 3 | 0.5718 |
| graca | **10** | **1.0000** | 0 | 0.0098 | 9 | 0.9997 | **10** | **1.0000** | **10** | **1.0000** | **10** | **1.0000** | **10** | **1.0000** |
| haplotype | **4** | **1.0000** | 0 | 0.3886 | 3 | 0.9931 | **4** | **1.0000** | **4** | **1.0000** | **4** | **1.0000** | **4** | **1.0000** |
| heinz | **18** | **0.6493** | 13 | 0.4754 | 15 | 0.6131 | 14 | 0.5739 | 14 | 0.5597 | **18** | 0.6487 | **18** | 0.6461 |
| kullmann | 2 | 0.8196 | 0 | 0.3577 | 2 | 0.8122 | 2 | 0.7301 | 1 | 0.6816 | 3 | 0.9817 | **4** | **1.0000** |
| logic-synt | 36 | 0.9986 | 12 | 0.8932 | 35 | 0.9459 | 24 | 0.9759 | 25 | 0.9781 | 36 | 0.9989 | **37** | **1.0000** |
| market-spl | **20** | **0.5500** | 11 | 0.2934 | 9 | 0.0125 | 9 | 0.1613 | 11 | 0.3431 | 11 | 0.2420 | 11 | 0.3022 |
| milp | **14** | **0.8235** | 2 | 0.2727 | 6 | 0.5556 | 9 | 0.6399 | 10 | 0.6964 | 4 | 0.5197 | 4 | 0.5389 |
| minlplib | 38 | 0.9972 | 9 | 0.9321 | 14 | 0.7229 | 19 | 0.9802 | 10 | 0.9612 | 36 | **0.9996** | 36 | 0.9994 |
| miplib | **46** | **0.8942** | 11 | 0.3900 | 20 | 0.6045 | 24 | 0.8046 | 21 | 0.7188 | 20 | 0.7024 | 23 | 0.7095 |
| mps | 1 | **1.0000** | **2** | **1.0000** | 0 | 0.8824 | **2** | **1.0000** | **2** | **1.0000** | **2** | **1.0000** | **2** | **1.0000** |
| oliveras | 47 | 0.7924 | 21 | 0.4639 | 47 | 0.8314 | **61** | **0.9983** | 59 | **0.9983** | 41 | 0.9773 | 41 | 0.9784 |
| pbfvmc-for | 10 | 0.9992 | 1 | 0.5562 | 0 | 0.5031 | 2 | 0.6082 | 3 | 0.5180 | **11** | **1.0000** | **11** | **1.0000** |
| poldner | **3** | **1.0000** | 2 | 0.9841 | **3** | **1.0000** | **3** | **1.0000** | **3** | **1.0000** | **3** | **1.0000** | **3** | **1.0000** |
| primes-dim | 63 | 0.8205 | 47 | 0.6871 | 58 | 0.7436 | 0 | 0.0000 | 62 | 0.8290 | 60 | 0.8073 | **63** | **0.8329** |
| radar | **6** | **1.0000** | 0 | 0.9743 | **6** | **1.0000** | 1 | 0.9827 | 3 | 0.9714 | **6** | **1.0000** | **6** | **1.0000** |
| random | 17 | **0.7727** | 1 | 0.5740 | 17 | **0.7727** | 17 | **0.7727** | 17 | **0.7727** | 8 | 0.7725 | 17 | **0.7727** |
| routing | **8** | **1.0000** | 5 | 0.6250 | **8** | **1.0000** | **8** | **1.0000** | **8** | **1.0000** | **8** | **1.0000** | **8** | **1.0000** |
| synthesis- | **5** | **1.0000** | 3 | 0.9745 | **5** | **1.0000** | **5** | **1.0000** | 3 | 0.9869 | **5** | **1.0000** | **5** | **1.0000** |
| trarea_ac | **5** | **1.0000** | 2 | 0.9822 | 3 | 0.9704 | 3 | 0.9669 | 3 | 0.9636 | **5** | **1.0000** | **5** | **1.0000** |
| ttp | 1 | 0.7154 | 1 | 0.7040 | 0 | 0.9296 | 2 | 0.9729 | 1 | 0.9594 | 2 | 0.9894 | **4** | **1.0000** |
| unibo | **8** | 0.4853 | 0 | 0.1142 | 5 | **0.4908** | 2 | 0.4002 | 2 | 0.3881 | 0 | 0.2980 | 0 | 0.3161 |
| vtxcov | 5 | 0.9995 | 0 | 0.9736 | 3 | 0.9986 | 0 | 0.9867 | 0 | 0.9711 | **8** | **1.0000** | **8** | **1.0000** |
| wnq | 1 | 0.9719 | 0 | 0.0000 | 0 | 0.9572 | 0 | 0.0000 | 0 | 0.3860 | 4 | 0.9895 | **7** | **0.9995** |
| Total | **1209** | **0.8834** | 665 | 0.6740 | 795 | 0.7548 | 931 | 0.8004 | 924 | 0.8398 | 1089 | 0.8653 | 1125 | 0.8674 |

Table 5: Multi-Solver Performance Comparison (By Dataset)