# OpenReview forum: "AutoPBO: LLM-powered Optimization for Local Search PBO Solvers"
_ICLR.cc/2026/Conference — Submitted to ICLR 2026_

### Official Review · Reviewer_5AuK · 2025-10-21

**Soundness:** 3
**Presentation:** 3
**Contribution:** 2
**Rating:** 6
**Confidence:** 4

**Summary:**

This paper introduces AutoPBO, a novel, closed-loop framework that leverages LLMs to automatically and iteratively optimize PBO local search solvers. The core of this framework is a multi-agent system that, combined with a greedy search, refines a modular solver named StructPBO. The framework demonstrates impressive empirical results, outperforming open-source competitors and showing performance competitive with the commercial solver Gurobi.

**Strengths:**

1. The paper presents a novel AutoPBO framework, which is capable of automatically and iteratively optimizing a PBO solver within a closed-loop, feedback-driven process.

2. The experimental results are strong: AutoPBO outperforms all evaluated open-source solvers and exhibits highly competitive performance against the top-tier commercial solver, Gurobi.

3. The paper includes an effective ablation study. The comparison between AutoPBO and its baseline, StructPBO, clearly demonstrates that the LLM-driven optimization framework provides a significant performance improvement .

**Weaknesses:**

1. The framework's generality may be limited. The authors note that LLMs struggle to optimize existing solvers directly, which necessitated the creation of the specialized StructPBO. This implies that applying AutoPBO to other existing PBO solvers might first require a significant refactoring effort to fit this modular structure.

2. The application of LLMs appears to be limited to prompt engineering.

**Questions:**

1. The framework employs a greedy algorithm to optimize functions sequentially (e.g., UpdateWeights before CalculateScore), citing the need to resolve function dependencies. Is this optimization order fixed, or is it automatically determined by the framework based on dependency analysis?

2. All benchmarks were split into training and testing sets using a 1:1 ratio. This ratio differs from more common practices. What was the rationale for choosing this specific 1:1 split?

**Details Of Ethics Concerns:**

N/A.

---

### Official Review · Reviewer_9gd3 · 2025-11-02

**Soundness:** 3
**Presentation:** 2
**Contribution:** 2
**Rating:** 4
**Confidence:** 3

**Summary:**

The paper proposes AutoPBO, a framework that uses a multi-agent, LLM-driven, feedback loop to iteratively improve the code of a local-search Pseudo-Boolean Optimization (PBO) solver. The authors first introduce StructPBO, a deliberately modular baseline whose key heuristics are exposed as functions (e.g., initialization, scoring, weight updates, escape moves). They then run an iterative process in which three LLM agents, Planner, Editor, and Evaluator, identify a function to modify, propose code changes, run the solver, and greedily keep the best-performing variant before moving to the next function. Experiments on PB16, MIPLIB, CRAFT, and a real-world dataset show consistent improvements over StructPBO and competitive performance versus six established solvers

**Strengths:**

1. Automating solver improvement with LLMs for general combinatorial optimization is important and under-explored.

2. The three-agent workflow and a clearly segmented baseline make LLM edits practical, testable, and rollback-safe. The framework design is easy to follow.

3. Across four benchmarks, AutoPBO improves over StructPBO and is competitive with strong baselines. Dataset-level plots/tables show a broadly non-negative trend

**Weaknesses:**

1. The entire pipeline is demonstrated with a single LLM. This leaves open whether (i) the method is model-agnostic, (ii) smaller/open models suffice, and (iii) gains scale with model size.
2. What did the LLM actually learn? We don’t see concrete code diffs or aggregated patterns of successful edits. Without this, it’s hard to judge whether the system discovers reusable heuristics versus dataset-specific tweaks.
3.  The score $\text{score}(x)=\alpha h\text{score}(x)+\beta o\text{score}(x)$ is clear, but the adaptation protocol for $(\alpha,\beta)$ and the penalty update mechanics remain narrative, not algorithmic

**Questions:**

1. How sensitive are results to the order of function updates? Can you report a random-order vs. dependency-aware comparison?
2. Have you observed cases where improving UpdateWeights hurt CalculateScore (or vice versa)? Would a keep-K best variants strategy per round mitigate such conflicts?
3. What are the LLM call counts, latency, and cost per full optimization loop? Any bottlenecks when scaling to larger solvers?
4. What happens if you train on two benchmarks and test on the third? Are gains durable under distribution shift?
5. Will you release code, prompts, seeds, and scripts sufficient to regenerate Tables 2–4?

I will consider changing my score if my concerns are resolved

---

### Official Review · Reviewer_mMgM · 2025-11-02

**Soundness:** 1
**Presentation:** 2
**Contribution:** 1
**Rating:** 2
**Confidence:** 4

**Summary:**

This paper proposes AutoPBO, a framework that leverages LLM to enhance heuristic strategies in local solvers. To further improve the guidance of the LLM, the authors introduce StructPBO, a structuralized local search PBO Solver designed to improve the performance of LLM designed for better heuristic. Experimental results on multiple benchmark datasets demonstrate that AutoPBO consistently outperforms classical solvers.

**Strengths:**

1. AutoPBO is the first to integrate LLM into pseudo-Boolean optimization, enabling automated heuristic enhancement and reducing reliance on manual expert tuning.
2. Experiments across four public benchmarks demonstrate that AutoPBO outperforms leading local search and complete solvers.

**Weaknesses:**

1. The paper lacks sufficient details of the experimental setup and necessary ablation studies.
2. AutoPBO employs three different agents combined with a greedy strategy to iteratively improve the heuristics generated by the LLM. The paper should provide a clearer explanation of how the framework operates, including how AutoPBO prevents failures such as code execution errors or hallucinated modifications.
3. The novelty and insights are limited. The overall idea appears to follow the same concept as FunSearch, with its main contribution being the adaptation to the PBO domain.

**Questions:**

1. This paper reports results only using DeepSeek R1. It would be more convincing if additional LLMs were included for comparison.
2. The authors should provide more implementation details, such as the number of LLM iterations required to discover heuristics, the complexity of the heuristics generated by the LLM, and the out-of-distribution performance, for example, applying the learned heuristic from one benchmark to another.
3. The paper would benefit from a deeper analysis of AutoPBO itself, including cases where AutoPBO fails, the roles and importance of different agents, and the contribution of StructPBO within the overall framework.

---

### Official Review · Reviewer_oXiR · 2025-11-03

**Soundness:** 2
**Presentation:** 3
**Contribution:** 2
**Rating:** 2
**Confidence:** 4

**Summary:**

This paper introduces AutoPBO, a novel framework that leverages Large Language Models (LLMs) to automatically enhance local search solvers for Pseudo-Boolean Optimization (PBO), which traditionally rely on heuristics requiring significant manual expert effort. The proposed system makes several key contributions: it employs a multi-agent LLM system (comprising a planner, editor, and evaluator) to iteratively generate, edit, and evaluate code modifications in a closed loop ; it introduces a new, modular solver called StructPBO, which is specifically designed to be easily understood and optimized by LLMs ; and it uses a greedy iterative strategy to progressively integrate the best-performing changes, managing complex code dependencies. Finally, extensive experiments on four public benchmarks show that AutoPBO significantly improves its baseline's performance and achieves results superior to state-of-the-art open-source solvers.

**Strengths:**

- The paper's originality comes from its application of LLMs to optimize PBO solvers, a previously unexplored area. The framework's design, which includes a multi-agent system and a new, modular StructPBO solver specifically designed for LLM modification, constitutes a novel approach to the problem.
- The paper demonstrates quality in its methodology. The multi-agent framework provides a complete, feedback-driven optimization loop. The empirical evaluation is thorough, covering four benchmarks and comparing against six SOTA competitors. The authors also performed parameter tuning for competitors, which strengthens the validity of the experimental comparisons.
- The work's significance lies in addressing the challenge of manual heuristic design in combinatorial optimization. By presenting an automated method, it suggests a new direction for solver development. The central finding, that an LLM-optimized solver can outperform existing open-source SOTA solvers and compete with a commercial one, is noteworthy and demonstrates the potential of LLMs in this complex domain.

**Weaknesses:**

- Lack of Insight into Discovered Heuristics: The paper validates its framework using performance metrics  but provides no scientific insight into the heuristics it automatically designed. The work is presented as a black box, making it unclear what algorithmic changes led to the improved scores. The authors must provide a qualitative analysis, including specific "before" and "after" code segments for key functions. This is necessary to explain the algorithmic changes and allow for independent testing and comparison of the new heuristics.
- Omission of Optimization Cost and Practicality: The paper completely omits any discussion of the computational cost required for the AutoPBO optimization loop. The described process—iteratively generating, compiling, and evaluating multiple solver versions in parallel on a training set—appears extremely resource-intensive. Without this information, it is impossible to assess the method's practicality or reproducibility. The authors should report the resources (e.g., number of LLM calls, total evaluation runs, or wall-clock time) required to run the optimization process for a typical benchmark.
- Limited Generality and High Refactoring Cost: The AutoPBO framework is not a general tool for optimizing existing PBO solvers. Its success is critically dependent on the pre-built, modular StructPBO solver, which was explicitly designed for this task. The authors themselves note that existing SOTA solvers are too complex and coupled for the LLM to optimize. This implies that applying this paradigm to any new solver or problem domain would first require a cumbersome and complex manual effort to refactor that solver into a similar "LLM-friendly" modular structure, significantly limiting the framework's practical generality and ease of adoption.

**Questions:**

- To address the lack of insight into the discovered heuristics , the authors must provide a qualitative analysis. This should include specific "before" and "after" code segments for key functions, explaining the algorithmic changes discovered by the LLM. Furthermore, the final, optimized AutoPBO source code must be made publicly available to allow for reproducibility and independent comparison.
- The paper must quantify the computational resources required for the AutoPBO optimization loop to assess its practicality. Please report the approximate number of LLM calls, the total number of solver-versions evaluated, and the total wall-clock time used during the optimization phase for a typical benchmark.
- Clarity is required regarding the framework's generality and the prerequisite human effort.

---

### Official Review · Reviewer_TPio · 2025-11-03

**Soundness:** 1
**Presentation:** 2
**Contribution:** 2
**Rating:** 2
**Confidence:** 4

**Summary:**

This paper introduces AutoPBO, a framework that leverages Large Language Models (LLMs) to automatically optimize local search solvers for Pseudo-Boolean Optimization (PBO) problems. The approach employs a multi-agent system consisting of three specialized agents: a Code Optimization Planner that identifies optimization opportunities, a Code Editor that implements modifications, and a Modification Evaluator that provides feedback. These agents work iteratively with a greedy search strategy to improve solver heuristics. The authors also propose StructPBO, a restructured baseline solver with clearer modular organization to facilitate LLM-based optimization. Experiments on four benchmarks (PB16, MIPLIB, CRAFT, Real-world) demonstrate improvements over baseline local search methods and competitive performance against state-of-the-art solvers including commercial solvers like Gurobi.

**Strengths:**

The paper's strength's:

Clear problem motivation and context for PBO solving
Figure 1 effectively illustrates the overall framework architecture
Good organization of related work and positioning

The application of LLM-based automated algorithm design is novel in addressing PBO solvers. Prior work (FunSearch, EoH, ReEvo, AutoSAT) has focused on simpler algorithms or specific problem types, while this work tackles general-form optimization problems with complex solver structures. The identification of this challenge and the proposed solution are valuable contributions.

The technical execution is competent but not exceptional. The multi-agent framework itself is relatively straightforward, essentially applying established LLM interaction patterns to a new domain. The more interesting contribution is StructPBO's design to facilitate LLM optimization.


The work demonstrates that LLM-based optimization can be effective for complex combinatorial optimization solvers, which could inspire similar approaches for other solver types. However, the practical impact is somewhat limited by modest performance gains over well-tuned baselines

The paper identifies and tackles the meaningful problem of applying LLM-based algorithm design to general-purpose solvers.

The evaluation spans 4 diverse benchmarks with 47 datasets and compares against 6 strong baselines including commercial solvers, providing good empirical coverage.

AutoPBO demonstrates consistent improvements over StructPBO across most datasets and achieves competitive performance with state-of-the-art methods, showing the approach has practical value.

**Weaknesses:**

The paper presents the following weaknesses:

The referenced variance analysis (Appendix A) is not provided
The multi-agent framework is fairly standard LLM orchestration without novel technical components
StructPBO's design is described as "following NuPBO" but the specific differences and design rationale are not detailed
Comparison primarily against self-created baseline (StructPBO) rather than direct optimization of existing solvers
The paper abruptly ends after experimental results with no conclusion section. This flaw leaves readers without: (1) synthesis of contributions, (2) discussion of limitations and when the approach fails, (3) broader implications, or (4) future research directions.

The paper reports using DeepSeek-R1 (600B+ parameter model requiring significant GPU resources) but describes a CPU-only system with two AMD EPYC 7763 processors and 1TB RAM. Running DeepSeek-R1 inference on this hardware configuration is technically infeasible. This raises questions about reproducibility

The paper uses wall-clock time budgets (60 seconds for training, 300 seconds for testing) rather than iteration-based or LLM-call-based budgets. API latency varies with network conditions, service load, and rate limiting. Results become non-reproducible across different setups or even different runs on the same setup. The fundamental unit of work in LLM-based optimization should be iterations/responses, not elapsed time, unless hardware and network are part of the optimization.

**Questions:**

How do you justify using time-based budgets given LLM response latency variability?
Can you report results using iteration counts (e.g., "10 LLM optimization rounds") rather than wall-clock time?
What specific design decisions went into StructPBO beyond "following NuPBO"? How much manual effort was required to create it?
Why were Conclusion and Future Work sections omitted?

**Details Of Ethics Concerns:**

This paper has a number of issues but I just want to say that there is no conclusion/future work. This was not ready for submission and I think should have been desk-rejected.

---

### Official Review · Reviewer_UMSc · 2025-11-04

**Soundness:** 1
**Presentation:** 3
**Contribution:** 1
**Rating:** 0
**Confidence:** 4

**Summary:**

The submission presents an LLM-based framework designed to automatically enhance pseudo-Boolean optimization (PBO) solvers that use local search. The motivation is that local search PBO solvers rely heavily on complex heuristics—such as scoring functions and weighting schemes—that require extensive expert tuning. The submission aims to automate this design process using large language models.
The proposed system is organized around a multi-agent architecture composed of three collaborating LLM-based agents that iteratively improve solver code:
1. Code Optimization Planner – This agent analyzes the existing solver code to identify which functions or heuristics can be improved. It generates modification plans by suggesting potential optimization directions.
2. Code Editor – Guided by the planner’s suggestions, this agent performs the actual code modifications. It rewrites relevant parts of the solver’s source code to implement the proposed changes, ensuring that function definitions remain valid and the solver remains executable. The editor then compiles and runs the modified solver to collect performance data.
3. Modification Evaluator – After the code is modified and executed, this agent analyzes the results. It evaluates whether the new solver version runs correctly and whether it improves performance metrics. The evaluator provides structured feedback to the Code Editor, which uses it to refine subsequent versions of the code.

These three agents operate within an iterative optimization loop. In each iteration, multiple modified solver versions are generated and evaluated. The framework then applies a greedy selection strategy to choose the best-performing version and uses it as the basis for the next iteration.
The LLM-based solver optimization is applied to a local search PBO solver, called StructPBO, that is designed to be easily comprehensible by LLMs. The optimized code, called AutoPBO, is then compared to StructPBO on multiple benchmark suits. Moreover, it is shown how AutoPBO performs compared to other state-of-the-art solvers.

**Strengths:**

- The paper is clearly written and well-structured, making it easy to follow. The authors effectively convey the high-level ideas and workflow of the AutoPBO framework, allowing readers to understand both the motivation and the core mechanisms behind the proposed multi-agent LLM optimization approach.
- The paper presents experimental results demonstrating that AutoPBO improves solver performance and outperforms the considered open-source state-of-the-art solvers across multiple benchmark suites.

**Weaknesses:**

- AutoPBO relies on a highly structured solver design (StructPBO), meaning it cannot be directly applied to existing complex solvers without significant re-engineering.
- The magnitude of improvement achieved by AutoPBO over StructPBO seems relatively modest. The reported performance gap between AutoPBO and StructPBO is comparable to the difference between AutoPBO and the commercial solver Gurobi, which the authors describe as “competitive”. This raises questions about how substantial the LLM-driven optimization gains actually are.
- The LLM-based optimization is evaluated on only one solver implementation, making it unclear how well the approach generalizes to solvers of, for example, lower quality.
- The paper lacks detailed technical descriptions of the StructPBO implementation and its constituent functions. Even in the appendix, the explanations remain high-level, preventing a thorough understanding of how the baseline solver operates.
- The code is not publicly available, and there is no reproducibility statement provided. As a result, the reported experiments cannot be independently verified, making the work not reproducible.
- There is no formal justification or theoretical analysis provided explaining why the proposed framework works well. The evaluation solely relies on experiments that cannot be reproduced.

**Questions:**

- How can the experiments be reproduced?
- Why does StructPBO perform better in the experiments than NuPBO if the former is based on the latter?
- StructPBO seems to perform well already compared to other solvers. What happens if a much worse initial solver is started with? Is there a possibility that at the end every input solver is modified to some similar output solver?
- Is there any formal justification for why LLM-powered optimization works particularly well for local search PBO solvers or why not to use LLMs to solve PBO directly?

**Details Of Ethics Concerns:**

There is no reproducibility statement included in the submission. There is neither a link to the proposed solver nor any concrete details on its implementation (not even in the appendix) provided. This lack of transparency raises concerns about the credibility and reproducibility of the experimental results.

---

### Official Review · Reviewer_Smn6 · 2025-11-06

**Soundness:** 3
**Presentation:** 2
**Contribution:** 2
**Rating:** 2
**Confidence:** 3

**Summary:**

In this paper, the authors proposed a multi-agent framework to solve pseudo-boolean optimization (PBO), which can be considered a special class of integer programs with binary variables. In the framework, there are three large language models (LLMs) acting as three agents with different roles. They evaluated their framework against existing solvers across different datasets.

My main concern of accepting this paper to a venue like ICLR is the lack of novelty. The multi-agent framework is not new, and the proposed framework of a planner, a worker (i.e., the code editor), and an evaluator (i.e., the modification evaluator) is quite standard. I am not sure it is tailored to solving PBO.

**Strengths:**

* The authors proposed a multi-agent framework to solve PBO, which is an important class of optimization problems.
* The authors evaluated the proposed framework and other methods across several benchmarks.

**Weaknesses:**

**Limited Novelty.** I think that using a multi-agent framework of a planner, an editor, and an evaluator is not novelty enough for a conference like ICLR. I would like to see some innovations made to address the challenges of solving PBO.

**Lack of Clarity and Details.** I think that the framework is not well described. This also leads to my comment on novelty; with the details provided in the paper, I do not find a significant innovation in the proposed framework.
* The prompts for different agents are not provided in the appendix.
* What are the functions generated by the LLMs? Some examples would be helpful to see the differences from the functions obtained from human heuristics.
* What are the differences between *AutoPBO* and *StructPBO*? I thought *AutoPBO* is what has been proposed, until the description of *StructPBO* in `Section 3.4`. The authors also compared these two in experimental results (`Table 2`, `Figure 2`, and `Figure 3`). So what are the relationship between these two?

**Performance on Benchmarks.** The proposed framework has a performance quite similar to the existing solver *NuPBO*, especially when looking at the detailed breakdown in `Table 5`. It is beaten by Gurobi, a popular commercial solver, in most benchmarks. I know that the inability to achieve state-of-the-art performance should not be a reason for rejection, *if* there are interesting insights in the paper. Unfortunately, I could not find enough novelty in the methodology to outweigh the relatively weak performance on benchmarks.

**Questions:**

Please see my major comments in "Weaknesses".

One additional question: Can all PB constraints be written in the normalized form with non-negative coefficients *and non-negative thresholds* (`Line 116`)? I understand that the signs of coefficients $a_j$ can be flipped by the negation of literals. But how to make negative thresholds non-negative?

---

### Official Review · Reviewer_Gy4R · 2025-11-07

**Soundness:** 2
**Presentation:** 2
**Contribution:** 2
**Rating:** 4
**Confidence:** 2

**Summary:**

This paper proposes AutoPBO, a multi-agent, LLM-driven framework for automatically optimizing local-search-based pseudo-Boolean optimization (PBO) solvers. The method uses three cooperating LLM agents (Planner, Editor, Evaluator) and a greedy iterative process to improve solver components such as scoring and weighting heuristics. It introduces a ''structuralized” baseline solver (StructPBO) designed for readability and modular optimization. Experiments on 4 benchmarks (PB16, MIPLIB, CRAFT, Real-world) show consistent improvements over StructPBO and competitive performance with established solvers (NuPBO, OraSLS, PBO-IHS, Gurobi, SCIP).

**Strengths:**

1. **Motivation.** The motivation is concrete: LLMs have shown promise for algorithm discovery, but not yet for general-form combinatorial solvers such as PBO. The authors make a credible case for the novelty of applying LLMs to heuristic tuning in such solvers. The distinction between “specific-problem evolutionary algorithms” and “general-structure solver optimization” is convincingly articulated.

2. **Sound modular design (StructPBO).** The introduction of StructPBO as a structured, decomposed solver (Algorithm 1) is an excellent engineering decision. It enables isolation of functions like *UpdateWeights, CalculateScore*, etc., making it possible for LLM agents to perform localized reasoning. The modular structure (Appendix B-C) is logically consistent and cleanly aligned with local search principles.

3. The greedy sequential optimization of solver functions (Section 3.3) is carefully justified with a dependency argument, especially the example explaining the interplay between *UpdateWeights* and *CalculateScore*.

4. **Empirical validation.** The experiments are comprehensive (47 datasets, 4 benchmarks). The evaluation metrics (#win, avg score) are standard in PBO research and clearly defined in Eq. (4.1.4). The per-dataset table (Appendix E) substantiates the claims of consistent improvement.

**Weaknesses:**

1. Although Section 3 defines three agents, there is no experiment isolating the effect of each. For example, it is unclear whether the Planner alone contributes most of the gains, or whether Evaluator feedback loops matter. A simple ablation (“No Evaluator”, “Single-agent baseline”) would strengthen causal claims.

2. In Section 3.3 (“Convergence to Optimal Solution”), the term “convergence to optimal” is misleading. The process is a greedy heuristic without any convergence guarantee. The text even says “progressively constructs the improved solver” but never formalises stopping criteria or potential oscillations if LLM modifications degrade prior gains. Maybe, clarify that “optimal” refers to empirical performance improvement, not theoretical optimality.

3. The formalization of the PBO preliminaries (Eq. 1–2) is largely correct, but a few inconsistencies exist. For example,
    (a) In the definition of $\text{(viol)}(c)$, the sum works because $l_j$s are $0/1$. Though in normalisation, authors assume that
         $a_j,b\in \mathbb{N}_{0}^{+}$. Yet, they later claim “without loss of generality”. That’s only true if coefficients can be negated
         consistently, which fails when constraints contain mixed signs. A short proof sketch should show the conversion explicitly.
    (b) $\text{smooth}(c)$ can be zero? Then definition of the penalty does not work.

4. In Line 16-17 of Algorithm 1, doesn't it terminate immediately after the first iteration if $\alpha*$ is non-empty, preventing full use of the cutoff time. Presumably this check should occur after the while-loop. Otherwise, the solver may exit prematurely after the first feasible α. This might be a typesetting indentation error but should be fixed.

5. The authors use DeepSeek-R1 LLM but don’t specify model size or inference cost. The total compute and number of optimization rounds are not reported. This omission hinders reproducibility and cost-benefit analysis. A single LLM iteration might take hours, and the total wall-clock cost of obtaining AutoPBO vs. StructPBO is critical.

Overall, the math is conceptually correct but lacks explicit formulaic rigor, but there are no derivations or theoretical results to verify beyond heuristic definitions.

**Questions:**

1. While the authors mention “variance analysis in Appendix A,” it’s not actually present (Appendix A only states LLM proofreading). There’s no evidence of statistical tests (e.g., Wilcoxon or t-tests) to confirm significance. The average score increases (e.g., $+0.011$) could fall within noise margins.
2.The framework is described as “closed-loop, feedback-driven,” but the feedback loop isn’t formalized. How is Evaluator feedback parsed or weighted?
3. Section 3.2 Table 1 includes “Strict JSON format adherence''. It is unclear to me whether this is actually required or just a design tip.
4. Appendix C lists “UpdateWeights: Heuristically selects the most promising variable”. Is it a copy-paste error (should adjust weights, not select variables).

---

### Official Review · Reviewer_LgvR · 2025-11-11

**Soundness:** 1
**Presentation:** 1
**Contribution:** 1
**Rating:** 2
**Confidence:** 4

**Summary:**

This paper introduces AutoPBO, a multi-agent system to improve local search solvers for pseudo-boolean optimization (PBO) problems. Each agent is driven by a large language model (LLM). The system iteratively optimizes submodules of an existing PBO local search solver to improve its heuristics. Empirical evaluations are provided on four benchmarks.

**Strengths:**

The idea of using LLM to enhance local search heuristic is interesting but the current version of the manuscript needs major rework.

**Weaknesses:**

1. The organization of the manuscript is confusing.
It introduces AutoPBO and StructPBO but it is unclear they are related. Is StructPBO the output of AutoPBO?
StructPBO is mentioned multiple times before its introduction in Section 3.4 so it seems the StructPBO section should be placed earlier.
Many sections lack details. For example, Algorithm 1 has many undefined terms such as $\Delta$Penalty (line 7 and 8). The seven independent functions do not have any details or explanations (even in Appendix C). Another example is lack of definition for h_score_ratio on line 214. Overall, I cannot fully understand the contributions of this paper so it would benefit from a major rewrite.

2. The empirical evaluations are hard to parse. Table 2 presents the results for AutoPBO vs StructPBO. My previous confusion continues on how these two differ. From Table 3, Gurobi performs better than AutoPBO so it is unclear what the significance of AutoPBO is. Furthermore, there is no time to best solution metric (there is a common cutoff time of 300 seconds) and including that along with objective values is important.

**Questions:**

1. The title for section 3.3 is "Convergence to Optimal Solution". I found this misleading as local search is fundamentally an incomplete solver. Does "optimal solution" here refer to something else?

---

### Meta-Review · Area_Chair_XHrW · 2026-01-07

**Summary:**

The paper introduces AutoPBO, a multi-agent LLM-based framework to automate the design of heuristics for Pseudo-Boolean Optimization (PBO). While the application of LLMs to combinatorial solver optimization is a timely topic, the submission has several fatal flaws. Most importantly, multiple reviewers and the Area Chair raised serious concerns regarding the reproducibility and credibility of the results, noting the absence of source code, implementation details, and even referenced appendices. Furthermore, the technical novelty is considered limited as it follows established multi-agent patterns, and the performance gains over state-of-the-art commercial solvers are not clearly demonstrated. Given the lack of author rebuttal to address these critical issues, I recommend to reject this paper.

**Reviewer Concerns:**

No rebuttal was submitted, and therefore none of the reviewers’ concerns were addressed.

**Reviewer Scores:**

All other reviewers did not explicitly indicate any intention to change their scores.

---

### Decision · Program_Chairs · 2026-01-26

Reject